# Use of Complementary and Alternative Medicine in Axial Spondyloarthritis: A Qualitative Exploration of Self-Management

**DOI:** 10.3390/jcm8050699

**Published:** 2019-05-17

**Authors:** Abbie Jordan, Hannah Family, Kelly Blaxall, Fiona M. Begen, Raj Sengupta

**Affiliations:** 1Department of Psychology, University of Bath, Bath BA2 7AY, UK; kellyblaxall@live.co.uk (K.B.); fiona.m.begen@bath.edu (F.M.B.); 2Department of Pharmacy and Pharmacology, University of Bath, Bath BA2 7AY, UK; h.e.family@bath.ac.uk; 3Royal National Hospital for Rheumatic Diseases, Bath BA1 1RL, UK; rajsengupta@nhs.net

**Keywords:** axial spondyloarthritis (axSpA), ankylosing spondylitis (AS), complementary and alternative medicine (CAM), self-management, self-regulatory model, qualitative, patients

## Abstract

Axial spondyloarthritis (axSpA) is an inflammatory rheumatic condition that is often subject to diagnostic delays. Individuals with axSpA report using complementary and alternative therapies prior to and following diagnosis, though little is known concerning reasons underlying use of such therapies. This study provides detailed insights into the motivation and experiences of complementary and alternative medicine use within a population of individuals with axSpA. Open-ended surveys were completed by 30 individuals (20–69 years; 17 females) diagnosed with axSpA. Subsequent telephone interviews were conducted with eight individuals (39–70 years; five females) diagnosed with axSpA. Data were analyzed using reflexive inductive thematic analysis. Themes of “a learning curve”, “barriers to complementary and alternative therapy use” and “complementary or mutually exclusive” illustrated how participants” increasing understanding of their condition empowered them to explore complementary and alternative therapies use as an adjunct to mainstream care. Individuals with axSpA recommended greater integration between mainstream and complementary and alternative therapies, valuing informed advice from mainstream healthcare professionals before selecting appropriate complementary and alternative therapies for potential use. Healthcare professionals should be proactive in discussing complementary and alternative therapy use with patients and supply them with details of organizations that can provide good-quality information.

## 1. Introduction

Axial spondyloarthritis (axSpA) is a chronic inflammatory rheumatic condition affecting the spine and peripheral joints, and is associated with ongoing pain, restricted movement and varying degrees of long-term disability [1]. AxSpA comprises non-radiographic axial spondyloarthritis (nr-axSpA) and ankylosing spondylitis (AS). This revised classification introduced in 2009 was to enable earlier diagnosis of the condition, previously typically taking 8–10 years. The classification of AS is reliant on the demonstration of changes on an X-ray, which is often apparent many years after the onset of symptoms. By encompassing changes on MRI which often occur earlier in the disease course, the nr-axSpA classification have been validated in this disease area. 

AxSpA represents a major financial burden for social and healthcare systems and has significant day-to-day implications for individuals diagnosed with the condition and their families [2,3]. Global prevalence of axSpA is estimated to be 0.1–1.4% [4] with an average age at onset of 17–24 years and a greater prevalence in men [2]. Distinguishing axSpA from more common mechanical back pain conditions can be challenging for non-specialist practitioners [5] and many individuals with axSpA experience delays of up to 9.8–10.4 years between initial symptom onset and formal diagnosis [6,7].

During the initial period between symptom onset and diagnosis, evidence from other rheumatic disease literature suggests that individuals attempt to self-manage their condition and seek to attribute symptoms to known, external factors such as over-exercise, prior injury, or life stressors. These misattributions are compounded by ignorance of symptoms associated with inflammatory conditions, and can lead to a delay in help-seeking [8]. It is only when misattributions and self-management strategies fail to explain and control the progression of symptoms over time that individuals are prompted to seek medical advice [8,9]. Where a credible diagnosis is not forthcoming, or symptoms are not adequately attenuated by medical intervention, individuals often turn to complementary and alternative medicine therapies (CAMs) to address ongoing difficulties associated with their condition [10]. CAMs represent a diverse group of therapies that are “not fully integrated into the mainstream healthcare system” of a particular country. While robust evidence supporting CAM’s efficacy in the treatment of rheumatic conditions is currently lacking [11], individuals report subjective benefits associated with CAM use [12] and advocate a more integrated approach between CAM and mainstream medicine [13]. 

Prior to diagnosis and throughout the course of axSpA, individuals experience several physical, emotional and psychosocial challenges that cannot be fully addressed by mainstream medicine alone [10,14]. Many of these individuals will never experience partial or full remission of their axSpA [15] and limited evidence suggests that they may pursue alternative treatment approaches as a result. In a UK-based study, 40% of participants with axSpA reported using at least one CAM therapy prior to their diagnosis [16] and data from Australian and Dutch studies report previous or current CAM use among post-diagnosis populations with axSpA ranges between 86–95% [13,17].

As yet, little is known about the reasons underlying CAM use in individuals with axSpA, the range of therapies that they access, or any benefits and negative outcomes that may be experienced in relation to their CAM use. This study aims to address this knowledge gap by using in-depth qualitative methods to explore the motivations, experiences, and outcomes of CAM use within a population of individuals with axSpA.

## 2. Experimental Section

### 2.1. Design

Adopting both a relativist epistemology and phenomenological approach, this study sought to gain a better understanding of how individuals with axSpA make sense of using CAM and seeking help from CAM practitioners to manage their axSpA. The study implemented an online qualitative survey design, with semi-structured telephone interviews completed with a sub-sample of participants after completion of the survey. Qualitative surveys and semi-structured interviews were implemented to enable a flexible and fluid structure, shaped by participants’ own experiences to produce situated knowledge [18]. 

### 2.2. Participants

Qualitative survey participants (*n* = 30) were recruited through an expression of interest provided at an ankylosing spondylitis (AS: type of axSpA) public engagement event or via dissemination of study information in an AS specific society newsletter and social media sites. When recruiting participants to the study, the term “AS” was used instead of “axSpA” to fit with participants’ description and understanding of the condition. Survey eligibility criteria required participants to be: (1) aged 18+ years and (2) diagnosed with axSpA by a rheumatologist. Following survey completion, 20 participants expressed an interest in taking part in in-depth interviews. Interview participants (*n* = 8) were selected to ensure that a wide range of experiences with different CAMs were represented. To be eligible participants had to be: (1) aged 18+ years, (2) diagnosed with axSpA by a rheumatologist and (3) possess experience of using at least one CAM in relation to their axSpA symptoms. CAM was defined as any therapy or treatment outside of mainstream health care. A sample size of between five and twenty-five was deemed to be methodologically appropriate with the analytical approach and the fact that follow up interview data was to be used as an adjunct to the survey data [19]. A deliberate effort was made to recruit both male and female participants to ensure representation of a wide range of experiences across the study.

### 2.3. Materials

#### 2.3.1. Qualitative Open-Ended Survey

Using an online survey tool, participants responded to open-ended questions relating to their axSpA and CAM use in their own words. In addition to demographic questions, a total of 14 questions enquired about participants’ axSpA symptoms and diagnosis, past and present CAM use, motivations underlying CAM use and CAM choices, and challenges encountered when using CAMs. Survey items included: “What do you feel might have sped up the process of diagnosis?”, “Can you tell us why you decided to participate in CAM therapy/therapies in relation to your AS?” and “How does using CAM therapies fit with other treatment you receive for your AS from mainstream healthcare professionals?”. Participants were advised to provide as much detail as possible in their responses and that text boxes would expand as they typed. For the purposes of the survey, the term “AS” was used instead of “axSpA” to fit with participants’ description and understanding of the condition. Full details of survey items are shown in Appendix A.

#### 2.3.2. Semi-Structured Interviews

A collaborative topic guide (summarized in Table 1) was created by authors, with content specifically informed by discussion with individuals with axSpA during an AS public engagement event. The topic guide explored the individual’s experience of receiving a diagnosis of axSpA and their perceptions and experiences of using CAM to manage symptoms of axSpA alongside mainstream care.

### 2.4. Procedure

All participants provided their informed consent for inclusion before they participated in the study. The study was conducted in accordance with the Declaration of Helsinki, and the protocol was approved by the College of Liberal Arts Ethics Committee of Bath Spa University (No specific project identification code provided by committee) and the Research Ethics Approval Committee for Health Ethics Committee of the University of Bath (REACH EP 15/16 205). Survey participants provided online consent prior to completing the survey, and interview participants provided written consent prior to and verbal consent at the time of interview. Interviews lasted 19–54 min, were audio-recorded, and transcribed verbatim. Survey and interview transcripts were anonymized and entered into QSR International’s NVivo 10 qualitative data analysis software [20]. All participants were provided with debrief information following completion. On completion, survey participants were entered into a prize draw for high street shopping voucher (£50) while interview participants were provided with a high street shopping voucher (£10).

### 2.5. Data Analysis

Survey and interview data were analyzed using reflexive inductive thematic analysis, following Braun and Clarke’s [21] six-phase process. Thematic analysis is a well-documented and widely used method in qualitative research. The benefit of this method, particularly this research, is that it is not limited to any one epistemological standpoint or tied to one method of data which enables a flexible approach to analyzing qualitative data across a range of research questions [22]. For this study, using thematic analysis allowed for the survey and interview data to be analyzed and interpreted together. Analytical phases in reflexive inductive thematic analysis comprise: (1) data familiarization (listening to the interview and reading the transcripts), (2) data-driven semantic coding between authors at different times (3) theming (reading coding reports, identifying the recurring patterns in the data) (4) reviewing/refining themes (5) writing descriptions of individual themes and their titles (6) writing-up the study. Issues pertaining to establishing quality in qualitative research were addressed. Firstly, analyses were discussed among authors over multiple meetings to ensure agreement about data interpretation. Secondly, trustworthiness was established through ensuring that quotations presented in results reflected exploration of a range of participant accounts and the generation of a clear audit trail across the analytical process [22,23]. Participants were assigned a pseudonym for the purpose of ensuring confidentiality of participants when reporting study findings [24].

## 3. Results

Details of participant demographics in survey and interview populations are shown in Table 2. 

### 3.1. Summary of CAM Use in Survey and Interview Samples

Table 3 below provides detailed information about CAM use in participants in both the survey and interview samples. This includes information pertaining to which CAMs were used, continued and stopped in addition to when CAMs were used in relation to receipt of a formal axSpA diagnosis.

Data from Table 3 highlights the use of CAM both prior to and following a diagnosis of axSpA, with 53.3% and 75.0%, respectively, of survey and interview participants continuing to use CAM to manage their axSpA symptoms. Regarding therapies, the most commonly used CAM were movement exercise related therapies which included hydrotherapy, Pilates, yoga, and Tai Chi. Interestingly, in contrast to other CAM, 100% of participants in both interview and survey samples reported continuing to use movement and exercise related therapies to support axSpA symptom management. Reasons for stopping CAM use were varied, with the most frequently reported reason as the perception of CAM as ineffective or worsening symptoms (16.7% survey; 62.5% interview samples).

### 3.2. Qualitative Analyses

Three themes were created from qualitative analyses of the survey and interview data; (1) a learning curve, (2) barriers to CAM use and (3) complementary or mutually exclusive? These themes are described below and supported by quotations across a range of participant accounts. 

#### 3.2.1. A Learning Curve

Individuals with axSpA endeavored to manage their condition using CAMs, both prior to and following diagnosis. Taking an active role in selecting CAMs for use enabled participants to position themselves as experts in their health condition and CAM treatment. Below, Tracey and Amy describe how they investigated CAM use prior to diagnosis as a potential source of symptom relief following consultation with doctors and failure of their own attempts to self-manage their condition. Using CAMs, they sought greater understanding of her symptoms and how she might better be able to help herself in managing them.
“[Prior to diagnosis] I did stretching and things and I saw lots of osteopaths, chiropractors. I just didn’t understand…why I was in such a bad way. I had acupuncture, all sorts of things, prior to the diagnosis. And now [following diagnosis] I don’t have any manipulation at all now.”*(Tracey: Interview)*

And
“Initially I did not know what the problem was, the doctors were apparently not interested in getting to the bottom of it so I felt that I had to “sort myself out” as it were…I honestly feel that for me [following diagnosis], [CAM] is part of a holistic approach to keeping my symptoms at bay.”*(Amy: Survey)*

Participants described their experiences with CAMs as providing short-lived symptom relief, no change in symptoms, or increased pain. Although participants were not yet able to fully understand or control symptoms, growing awareness of their bodies and associated functioning allowed them to judge the effectiveness of a therapy; and importantly when it might potentially be harmful to their condition.
“I went to an osteopath…who kept pulling my neck out and yanking it from one way to the other, and on the third session I couldn’t take it anymore, so I was crying. So, I decided to stop that.”*(Hazel: Interview)*

Simultaneously, participants continued to pursue a mainstream diagnosis for their condition. For Matthew, consultation with a CAM practitioner contributed directly to the process of obtaining a diagnosis of axSpA.
“I booked to see [chiropractor] and during the consultation I gave him a detailed account about my medical history. It was at this point that he suggested that I may be suffering from AS [axSpA] and actually wrote a letter to my GP recommending that I be blood tested for this. Within 6 months I had the diagnosis and was receiving the appropriate medication.”*(Matthew: Survey)*

For most participants, obtaining a mainstream diagnosis and appropriate treatment for their symptoms was problematic, and many experienced significant delays which impacted on their longer-term health outcomes. Participants identified factors which would have contributed to a quicker axSpA diagnosis. Many had encountered a fundamental lack of understanding of axSpA in primary care level and beyond, and recommended improvements in knowledge about the condition. For example, Andrew called for:
“Recognition, awareness, understanding (through education) of the condition fundamentally at primary care level (GP) as well as at Hospital Consultant level, including cross-disciplinary professions. Having… been diagnosed by a Consultant Ophthalmologist with iritis at the age of 13 (related autoimmune condition) but not for some time mentioned and linked to AS [axSpA].”*(Andrew: Survey)*

Participants also expressed the need for healthcare professionals to appreciate the diverse range of individuals who can be diagnosed with axSpA. For example, Amy explained that as a young woman who was able to engage with paid work, she perceived that her symptoms were addressed healthcare professionals. This may well be due to Amy not being recognized as a “typical” patient with axSpA. Amy explained that:
“I think because I was a young woman… also, as I was still able to work (albeit in great discomfort) it was not deemed chronic enough to warrant further study. Once an initial referral to a physiotherapist… ran its three-month course and no change for the better I was just signed off with a neural suppressant prescription (this didn’t work).”*(Amy: Survey)*

Healthcare professionals’ lack of knowledge about axSpA had substantial impact on participants beyond the symptoms of the condition itself. Participants described how they did not feel “listened to” or “believed” about their symptoms, and this led to feelings of alienation on the part of some participants who began to question their own instincts and understanding of their bodies.
“I just feel that they should have listened to me more, I should have been listened to... Because they did every test imaginable and couldn’t find anything wrong with me, just to be brushed off, you know. As I say…I knew my body and I knew that this was more than just a back pain.”*(Julia: Interview)*

Consequently, a minority of participants felt a sense of helplessness and withdrew from the diagnostic process entirely. With hindsight, they regretted the fact that they had not pursued an accurate diagnosis for their symptoms more fully at an earlier stage.
“A doctor recognizing the symptoms and not assuming it was mechanical: a need to lose weight. Also, I should have persevered. I felt fobbed off and just resigned myself to living with back pain.”*(Steve: Survey)*

For many participants, diagnosis of axSpA resulted in a sense of empowerment which facilitated information-seeking and exploration of CAM approaches that could be explored alongside mainstream treatment. Participants endeavored to equip themselves with the knowledge to better manage and become experts in their condition. Over time, they were able to identify CAMs that were most appropriate for specific symptoms.
“…hydrotherapy, that helps with all the stretching, which is well-documented is what AS patients need because they tighten up. So that’s good...Acupuncture, definitely, when there’s a specific symptom that might have developed that would help. It takes a bit of time, but that reduces the pain in that specific area.”*(Penelope: Interview)*

As participants learned more about their condition, understanding of their physical abilities grew, and they were able to monitor their day-to-day well-being and recognize when the need for further intervention was appropriate. Participants described how use of CAM enabled them to identify distinct time-points where they could, and did, take action to address poorly managed symptoms.
“I do a Pilates class every week and do some Pilates exercises the rest of the time. I…had a one-to-one teacher, but I now am part of the general class because I feel a lot more confident about…what I can do and what I can’t do. That’s made a huge difference with my flexibility and strengthening…But it’s also for me an alert system, so if there are things that I usually do that I’m beginning to find more difficult…then it enables me to do something else about it. I either do some more exercises or…I will go to an osteopath.”*(Amelia: Interview)*

For some participants, CAMs also represented an opportunity for self-management and a sense of control over their mainstream treatment by enabling them to avoid medications to which they experienced unwanted side effects, or to delay the introduction of medications which they were not yet ready to engage with.
“Nothing else was working and I have a low tolerance to opiate painkillers (i.e. they make me feel very sick)... Am reluctant to try anti Tumour necrosis factor alpha (TNF) because of the side effects (consultant supportive of this). Started Pilates because the stiffness was becoming incapacitating and this has helped enormously.”*(Helen: Survey)*

Participants also appreciated the more holistic approach offered by CAMs, and the impact that this could have on specific symptoms, and more widely on their general well-being and axSpA self-management. Below, Kyle and Liz explain how use of CAMs aided relaxation and sleep which helped to address outstanding pain issues.
“[Aromatherapy helps] Basically by relaxing me…so this pain I’ve got in my neck at the moment, my head has become rigid because of the pain. So, if you’ve got something which can relax you a little bit and make you even fall asleep…then you’re not feeling the pain.”*(Kyle: Interview)*

And
“I have never found any CAM treatment that is amazingly effective for pain, but the hot stone massage helps enough that if my pain is really bad or I feel particularly tense, I will book a massage.”*(Liz: Survey)*

#### 3.2.2. Barriers to CAM Use

Participants discussed the barriers that they encounter when considering and attempting to access CAMs. It was rare for participants to encounter a CAM therapist who had prior knowledge of axSpA, and this was a cause of concern for many. Levels of trust increased where practitioners were seen to make proactive attempts to further their knowledge of axSpA. As experts in their own condition, participants used their personal experience and available resources to facilitate practitioner learning and maximize understanding of axSpA prior to commencing treatment.
“…when I went for the Pilates assessment, she asked me would I get a letter off my doctor to authorise that they were happy for me to do this...she didn’t have a great knowledge of my condition…I took her a [axSpA-specific organization] patient handbook at the time…That always scares me a little bit, because…one of the greatest things is lack of awareness.”*(Julia: Interview)*

And
“Don’t expect [CAM practitioners] to know about AS [axSpA] and interview them when you go to your first appointment rather than they interviewing you! If they believe their CAM is a “fix all” then be extremely cautious as the best it can do…is assist with some pain or mobility.”*(Leanne: Survey)*

Participants were encouraged where practitioners showed an open and honest approach which presented a realistic view of treatment outcomes and facilitated participant autonomy in choosing to take part in CAMs.
“[Chiropractor was] always available to answer any questions…they never made any promises to say “you’ll think it’s like a miracle cure to you” or anything…it was just “if you’re happy to be treated then we’re happy treating you”*(Matthew: Interview)*

On a practical level, participants cited “time” and “cost” as barriers to initiating or continuing CAM use. For individuals in employment, the scheduling of CAMs during the day meant that they felt unable to attend these activities or were concerned that their requests for time off work to attend CAMs may be misunderstood by their employers.
“Hydrotherapy is fantastic but finding the time to take out of your working day to go and use it, because they’re often not open in the evening, or when you do go if you go to an evening session it’s full of people…that puts me off.”*(Penelope: Interview)*

And
“An employer may see a request to visit [CAMs] differently—a Doctor/Surgery/Hospital visit may be seen more as a ‘right’ and requirement… [CAM may] not being viewed as mainstream and essential healthcare and…seen with skepticism.”*(Andrew: Survey)*

For others, the cost of CAM was financially prohibitive, and freely accessible CAM was considered to potentially reduce NHS medication costs in the long term.
“It would be nice if a lot of them [CAMs] were considered to be funded on the NHS…because if…I could get funded to use a weekly hydro[therapy] pool, they’d save a lot of money because I wouldn’t have to take so many co-codamol…they wouldn’t be paying out so much on medications, would they?”*(Julia: Interview)*

“Context” of the CAM therapy was also a key factor for participants, and non-axSpA-specific CAMs were problematic for some individuals. Difficulties with body image and a perceived inability to fully take part meant that some participants felt alienated by these activities.
“I don’t really like them [other people in the class] looking at me…I’m very conscious of my shape now, I try and wear clothing that doesn’t show it off. So, at the moment I haven’t [attended classes].”*(Hazel: Interview)*

#### 3.2.3. Complementary or Mutually Exclusive

Despite highlighting distinctions between mainstream care and CAM, participants described the importance of adopting an integrated treatment approach. While many participants perceived treatments to be complementary rather than mutually exclusive, they considered these hierarchically; with priority being given to mainstream treatments and information obtained from medical sources in the first instance.
“CAM practitioners always told me to continue with my NHS treatments while receiving the holistic treatment. There was never any suggestion that my NHS treatment was "harming" me in any way (other than what everybody knows about unwanted drug side effects).”*(Robert: Survey)*

And
“Talk to their consultant or physiotherapist first. You have to be very careful, because if you get involved with a wrong one [CAM practitioner] or somebody doesn’t know what they’re doing, it can cause more damage than good.”*(Amelia: Interview)*

Participants shared the view that evidence-based CAMs should be recognized as a tool to manage axSpA symptoms and that greater awareness of CAMs among mainstream healthcare practitioners would be welcome. Several participants found that mainstream practitioners (especially doctors), were reluctant to discuss CAMs or the availability of therapeutic facilities outside the confines of their immediate hospital environment.
“The consultant that I saw…just told me to go swimming…but he didn’t tell me about hydrotherapy…I only found out about this facility, which is at the orthopedic hospital, through a free magazine that came through the post.”*(Tracey: Interview)*

While many participants argued for a more integrative view of CAM and mainstream healthcare treatments for axSpA, for some, this distinction was dependent on the specific nature of the CAMs and whether they were compatible with an axSpA diagnosis.
“It wouldn’t be contradictory to do acupuncture and hydrotherapy, but I think it would be contradictory to use a chiropractor or an osteopath in our condition from what the healthcare professionals are telling us.”*(Penelope: Interview)*

Participants reported that the boundaries between mainstream care and CAM were blurred when considering some therapeutic approaches. For example, in movement and exercise therapies, physiotherapy, hydrotherapy and Pilates were seen as a continuum of therapies which contributed to maintenance of participants’ flexibility and mobility. This conflation was unsurprising given that participants had often taken part in multidisciplinary axSpA-specific NHS-funded courses or exercise therapy sessions run by axSpA support charities; each of which were primarily led by physiotherapists.
“Hospital was excellent in offering physio and Pilates course, so you knew the basic way of doing [exercises] correctly.”*(Catherine: Survey)*

These more integrated approaches were popular with participants because they allowed “safe” exploration of CAM approaches within an environment supported by specialist axSpA healthcare professionals. 

## 4. Discussion

This study provides detailed insights into the motivations, experiences, and outcomes of CAM use within a population of individuals with axSpA through the 3 themes of “a learning curve”, “barriers to CAM use” and “complementary or mutually exclusive”. Participants revealed the experiential processes underpinning their selection of specific CAM therapies over the course of their axSpA; the subtle factors which enhanced or undermined their confidence in CAM therapy use; and the potential benefits of a more holistic approach to axSpA management which integrates mainstream and CAM therapies.

In line with literature suggesting a substantial delay in diagnosis of axSpA [6,7], participants described their early experiences in pursuing a credible diagnosis to explain their symptoms. Most of these participants attributed their diagnostic delay to a fundamental lack of knowledge and recognition of axSpA symptoms on the part of primary healthcare professionals. Although many participants investigated CAM therapies as an avenue to ameliorate and better understand symptoms prior to diagnosis; for others, the use of CAM therapies were instigated following axSpA diagnosis and in conjunction with mainstream pharmacological and therapeutic treatments. In accordance with the self-regulatory model [25,26], participants used their growing understanding of axSpA (illness representations) to inform selection of CAM therapies which were most likely to meet their outstanding therapeutic needs (coping responses), and to assess the efficacy of these therapies through trial and error (coping appraisals). Over time, as participants developed their expertise in axSpA self-management, they selected CAM therapies to maintain, expand, and monitor ongoing symptoms and functioning. At their best, CAM therapies contributed in normalizing participants’ day-to-day lives. Participants were empowered to take on new challenges and used additional CAM modalities to address symptom flares which disrupted everyday activities. 

In the current study, most of the survey and interview participants selected movement/exercise therapies for regular use in managing their axSpA. These choices were important given relatively strong evidence linking exercise with inflammation control and improved functioning in axSpA and other inflammatory rheumatic diseases [27,28]. Some participants indicated that they were aware of this connection based on their own information-seeking efforts and advice provided by mainstream health practitioners including rheumatologists or physiotherapists. It is noteworthy to consider that the empowerment and expertise exhibited by these participants did not develop within a vacuum. Mainstream medicine has a crucial role to play in educating and supporting patients to be active in making informed choices about how best to manage their axSpA [29,30,31,32]. Although some participants preferred to defer to “expert” mainstream practitioners regarding CAM choices in the first instance, the recent rise in patient-centered “shared decision-making” within mainstream rheumatology departments is likely to have played a crucial, if indirect, role in facilitating participants’ autonomy, and their subsequent investigation and engagement with suitable CAM therapies. 

In terms of maximizing their self-management, participants viewed CAM therapies as truly “complementary” to mainstream treatments and encouraged greater integration between the two. Prior literature suggests that individuals are unlikely to inform mainstream health professionals about their CAM use [13,33,34], and this can present a major challenge where CAM therapies (e.g., supplements or herbal remedies) interact poorly with conventional medications [35]. Given that many of these patients did not volunteer information on their CAM use because “the doctor never asked” [33], it is reasonable to conclude that such enquiries should become an integral part of rheumatology consultations. Although participants in the current study perceived some reticence on the part of their healthcare professionals in relation to CAMs; many participants reported that they would discuss any potential CAM use with these professionals prior to commencement. This is contrary to previously cited research, but in line with findings reporting that rheumatology patients are more likely to discuss CAM use where healthcare professionals exhibit a participatory decision-making style [36,37]. It is likely that the latter was the case for our study population because the majority had taken part in more holistic, multidisciplinary rehabilitation courses run by mainstream health services or national charities. 

CAM practitioners’ knowledge of axSpA was a key concern for participants, and CAM practitioners state that they are interested in collaborative working with mainstream healthcare providers [5]. Although specific CAM use is largely driven by individual patients rather than practitioners [17], the inclusion of evidence-based CAM modalities within mainstream multidisciplinary teams is already occurring in the UK. For example, the use of hydrotherapy in axSpA rehabilitation is both recommended [38] and available to many patients with axSpA on a short-term basis [39]. Accommodating the longer-term self-management need for such therapies within mainstream healthcare settings while additionally including a variety of other CAMs may prove more of a challenge however; particularly in light of wider population demands on finite healthcare resources. In axSpA and other inflammatory rheumatic diseases, national and European guidance suggests that mainstream healthcare practitioners should provide information regarding the limited evidence associated with the long-term efficacy of CAM therapies and should direct patients to trusted charities and organizations for information on this and other aspects of their disease [40,41]. It is likely that such organizations may be best placed to supply the guidance and support that many individuals with axSpA require when investigating the potential of CAM therapies to contribute to self-management of their condition.

In recognizing the insights gained through in-depth analysis of CAM use among individuals with axSpA, we also acknowledge the study’s limitations. Firstly, we note that there are differences regarding participation in the study according to gender. In both cases, more women compared with men, participated in the interviews (56.7%) and survey (62.5%). While it is important to be mindful of this when considering the study findings, such a gender divide with regard to CAM use is consistent with the wider CAM literature which shows a greater use of CAM among women compared with men in both general [42] and chronic ill health populations [37,43]. Additionally, as previously discussed, taking part in axSpA rehabilitation programs is likely to have impacted on participants’ self-management skills and potential CAM use. It would be interesting to recruit a similar group of individuals with axSpA who had not completed such courses, to compare their views on CAM therapies. Similarly, participants were recruited based on having used a CAM therapy in relation to their axSpA. Individuals with axSpA who have not used CAMs would likely express different views, which are worthy of comparison. Furthermore, participants within the study had been managing their axSpA for several years. An additional limitation concerns the range of CAMs considered in this paper. As the study involved asking participants to discuss their experiences of use of CAMs, data relied only on CAMs used and did not consider an exhaustive list of CAMs. Subsequent research could usefully focus on exploring how a wider range of CAMs are experienced by individuals with axSpA in terms of managing their axSpA related symptoms and effectiveness of specific CAMs. Further research is necessary to elicit the views of individuals over the course of their disease progression, to gain wider understanding of potential CAM use at different stages of axSpA development.

## 5. Conclusions

By using qualitative methods to gain in-depth insights into the lived experiences of CAM use in individuals with axSpA, this study has important implications for rheumatology practice. Through increased understanding of their condition, participants became active in disease self-management and were empowered to explore and use CAMs to complement mainstream medical treatments. CAMs that aided flexibility and that could be used to manage intermittent symptoms were favored. Participants reported that knowledge of axSpA among CAM practitioners was limited, and that they would value an integrated, holistic approach in which informed advice from mainstream health practitioners could aid them in selecting CAMs which might be appropriate for management of their axSpA symptoms. Mainstream medical professionals should be proactive in discussing CAM use with their patients and should provide them with details of trusted charities that offer further information and potential therapeutic resources.

## Figures and Tables

**Table 1 jcm-08-00699-t001:** Summary of topic guide ^a^.

Summary Of Topic Guide
Axial Spondyloarthritis (axSpA) development and diagnosis.
Feelings about diagnosis.
Complementary and alternative medicine therapy (CAM) use and changes in use over time.
The typical CAM consultation.
How CAMs help axSpA symptoms.
Current mainstream medical treatments.
CAM use and mainstream care.
Difficulties in CAM use.
Advice about CAM use for other individuals with axSpA.

^a^ See Appendix B for the full interview topic guide.

**Table 2 jcm-08-00699-t002:** Demographic characteristics of survey and interview participants.

Variable	Survey (*n* = 30)	Interview (*n* = 8)
Gender: *n* (%)		
Men *n* (%)	13 (43.3)	3 (37.5)
Women *n* (%)	17 (56.7)	5 (62.5)
Age (years) ^a^		
Mean (SD)	47.3 (12.4)	56.0 (13.0)
Range	20–69	39–70
Education level: *n* (%)		
Left school with no qualifications	3 (10.0)	Not reported ^b^
Apprenticeship	1 (3.3)	Not reported ^b^
School/college qualifications	10 (30.0)	Not reported ^b^
Undergraduate degree or equivalent	7 (23.3)	Not reported ^b^
Postgraduate degree or equivalent	9 (30.0)	Not reported ^b^
Currently in work: *n* (%)		
Yes	24 (80.0)	Not reported ^b^
No	6 (20.0)	Not reported ^b^
Occupation: *n* (%)		
Higher or intermediate managerial, administrative, or professional	7 (23.3)	Not reported ^b^
Supervisory, clerical and junior managerial, administrative, or professional	9 (30.0)	Not reported ^b^
Skilled manual worker with training or apprenticeship	4 (13.3)	Not reported ^b^
Semi-skilled or unskilled manual worker	1 (3.3)	Not reported ^b^
Casual Worker	0 (0.0)	Not reported ^b^
Unemployed/Currently not working	3 (10.0)	Not reported ^b^
Retired (reliant solely on state pension)	3 (10.0)	Not reported ^b^
Full-time student	1 (3.3)	Not reported ^b^

^a^ Two participants responded “years” and did not provide specific information, ^b^ Detailed demographic information about work and educational history was not collected from interview participants.

**Table 3 jcm-08-00699-t003:** Complementary and alternative medicine therapies (CAM) use in survey and interview participants.

Variable	Survey (*n* = 30)	Interview (*n* = 8)
Age at symptom onset (years)		
Mean (SD)	26.8 (11.9)	19.4 (5.6)
Range	10–58	12–30
Delay to diagnosis (years)		
Mean (SD)	8.8 (6.2)	13.6 (6.2)
Range	<1–25 ^a^	3–25
Previous CAM use to manage axSpA symptoms: *n* (%)	26 (86.7)	8 (100)
Current CAM use to manage axSpA symptoms: *n* (%)	16 (53.3)	6 (75.0)
Started CAM use: *n* (%)		
Before diagnosis	17 (56.7)	4 (50.0)
After diagnosis	9 (30.0)	4 (50.0)
Types of CAM used and stopped ^a^: *n* (%)		
Movement/exercise therapies-e.g., hydrotherapy, Pilates, Tai Chi, yoga	19 (63.3)	6 (75.0)
Stopped using exercise therapies ^b^	0 (0)	0 (0)
Manipulation/touch therapies, e.g., massage, chiropractic, osteopathy	17 (56.7)	5 (62.5)
Stopped using manipulation/touch therapies ^b^	8 (47.1)	5 (62.5)
Acupuncture	21 (70.0)	4 (50.0)
Stopped using acupuncture ^b^	12 (57.1)	2 (25.0)
Reasons for stopping use of CAM to manage axSpA symptoms		
Incompatible with current management of condition	2 (6.7)	1 (12.5)
Perceived to be ineffective or to worsen symptoms	5 (16.7)	5 (62.5)
Cost	1 (3.3)	1 (12.5)
No current need	1 (3.3)	1 (12.5)
Therapist change	1 (3.3)	0 (0)

^a^ Past and/or current CAM use regardless of perceived efficacy, ^b^ Percentage calculated for entire interview and survey samples rather than only those participants who indicated use of that particular CAM.

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
