# Peer review of "Use of Complementary and Alternative Medicine in Axial Spondyloarthritis: A Qualitative Exploration of Self-Management"

_jcm, 2019, doi:10.3390/jcm8050699_

Reviewer 1 Report

A holistic approach to the patient including CAM procedures is of great importance in the treatment of patients with chronic diseases such as axSpA.
The work is interesting, but requires several additions.

There is a  lack of data on which type of axSpA was diagnosed in the examined patients and why women predominate in the study group.
Ankylosing spondylitis (AS) occurs 2-3 times more often in a man. In contrast, psoriatic arthritis (PsA) - is more common in women. Women predominate in the study group (56.7%), so determining the type of axSpA is important.
The authors should also  discuss the fact that the questionnaires were more often answered by women. Is it due to the type of disease?, due to education level ?, or maybe women are more likely than men to use CAM?
I believe that after these corrections, the publication will be a source of knowledge for medical professionals about what patients on axSpA expect from doctors in the disease management.

Reviewer 2 Report

This is a very important manuscript because there is a growing interest in complementary and alternative medicine (CAM) across the patient population and medical community needs to understand the prevalence of CAM use within social contexts. 

There are some limitations in the study that should be mentioned in the manuscript:

- some of the CAM therapies may not have been included,

- the thematic areas detected could change if another dataset  was considered.
